# Deep Learning Based Partial Annotation Framework for Instance Segmentation in Histopathology Images

**Elad Arbel**                                                ELAD.ARBEL@AGILENT.COM

**Itay Remer**                                                ITAY.REMER@AGILENT.COM

**Amir Ben-Dor**                                              AMIR_BEN-DOR@AGILENT.COM

*Agilent Research Laboratories, Petach Tikva, Israel*

## Abstract

The applicability of Deep Learning based methods to image segmentation tasks in general, and to nuclei segmentation in particular, is currently limited by the effort required to collect large enough high-quality training data. In this work we describe a novel deep-learning based approach for instance segmentation that utilizes an easier to collect weak training data. Namely, a mixture of fully segmented nuclei and nuclei with only center locations specified. We demonstrate the robustness of our proposed approach - using 30% of fully segmented nuclei decreases the algorithm performance by only 4%.

**Keywords:** Fully Convolutional Neural Network, Weak Annotation, Instance Segmentation

## 1. Introduction

In recent years, digital pathology is gaining more popularly as many stained tissue-slides are digitally scanned with high resolution and viewed as whole slide images (WSIs) using PC monitors instead of standard microscopes. *Instance Segmentation* is the problem of identifying and delineating all instances of objects in an image. A key example of such task is nuclei segmentation in digital histopathology microscopy images where all nuclei need to be segmented. This task is a key step in many digital pathology analyses such as cell classification and various cancer grading tasks. Developing a robust nuclei segmentation method is particularly challenging due to the vast diversity of nuclei shape, color, orientation and density in different tissue and stain types (Kumar et al., 2017). State of the art algorithms for nuclei segmentation are dominated by Deep Learning (DL) based models (CPM, 2018; MoNuSeg, 2018). However, the performance of such models crucially depends on the size and quality of the available ground truth data that is used to train the model. Collecting this ground truth data is particularly challenging as exact pixel-level boundaries for thousands of nuclei need to be specified, by a domain expert. This annotation effort is extremely tedious and is the limiting factor to a broader applicability of DL models for nuclei segmentation. The relative ease of collecting partial annotation with only nuclei center location are recorded will allow large training dataset to be collected. This, in turn, makes this proposed mixed annotation strategy particularly appealing for nuclei segmentation tasks – training an DL model for a hard task (instance segmentation) using a combination of a small fully annotated dataset, and a large partially annotated dataset. We note that while mixed annotation approaches were used for DL model training in other settings (Shah et al., 2018), we are not aware of such approach being used for instance segmentation problem.

## 2. Methods

### 2.1. Dual-regression deep neural network for nuclei segmentation

Our approach is composed of three main steps (detailed below, see also Figure 1). **Ground truth encoding:** For each train image we have an associated ground truth segmentation of the pixels into non-overlapping nuclei. We compute for each nucleus its centroid, and two distance measures for each pixel. (i) a distance to the nearest nucleus centroid and (ii) a distance to the nearest nucleus

edge [Figure 1(c-d)]. Following the approach in (Kainz et al., 2015), we apply inverse exponential transform to these distances. **Network architecture:** We designed a variation of u-net architecture (Ronneberger et al., 2015), replacing the last classification layer with a dual-regression layer to predict the ground-truth surface maps. As a loss function we use a weighted sum of squared differences between encoded ground truth and model output. Roughly speaking, using a dual-layer as the network output enforces the model to implicitly learn a correlation between the two data-channels. **Post-Processing:** To convert the output network surfaces [Figure 1(e-f)] to nuclei segmentation we find foreground and background markers [Figure 1(g)], and apply markers-controlled watershed algorithm on the predicted edge surface. An earlier version of this architecture achieved a competitive segmentation score (AJI) of 0.62 (MoNuSeg, 2018).

## 2.2. Adapting partial annotations for training a deep neural network

We take advantage of the implicit separation between the two surfaces (nuclei detection and boundary detection respectively), and mask-out all partially annotated nuclei in the boundary channel during the training process. Specifically, we employ a boundary-mask around partially annotated nuclei (black pixels in Figure 1(d)), therefore, those specific nuclei are not scored for boundary prediction during the training process, but only scored for nuclei centers.

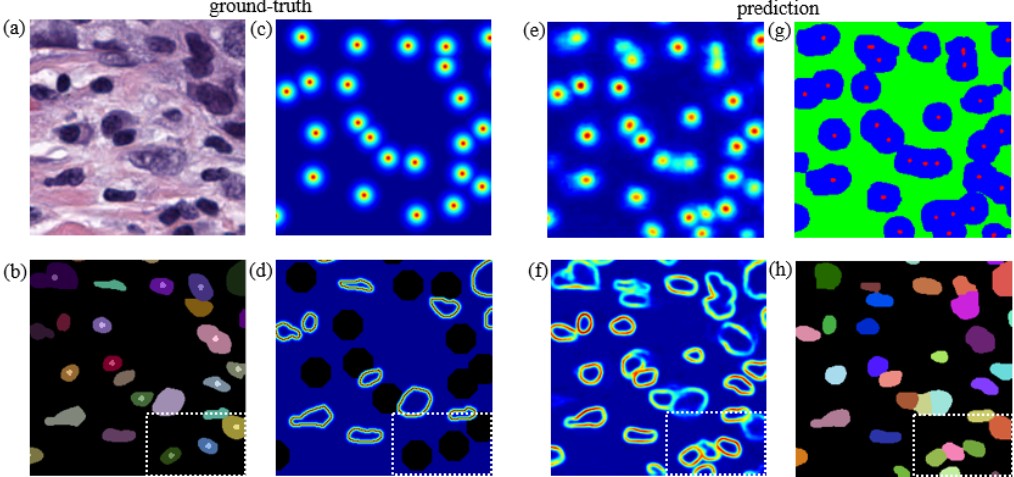

Figure 1: (a) RGB image (b) Ground truth segmentation image, white spots mark nuclei selected to be partially annotated (c) Nearest nuclei center distance transform (d) Nearest nuclei boundary distance transform. Partially annotated nuclei are masked (black pixels) (e) Network prediction of nuclei center distance transform (f) Network prediction of nearest nuclei boundary distance transform (g) Foreground and background markers in red and green colors, respectively (h) Instance segmentation prediction. White box (bottom row) highlights weakly annotated nuclei that were segmented correctly and a nucleus that was missed in the original annotation.

## 3. Experiments and Results

For training the model we used H&E stained multi organ nuclei segmentation datasets (CPM, 2018; MoNuSeg, 2018). For the evaluation described below, we have selected 14 out of the 40 images to be used as a test set. The remaining 26 images were used to train the model. Our model has reached AJI score of 0.57 which outperforms other DL approaches being evaluated on this test dataset

(0.51 and 0.56 for (Kumar et al., 2017) and (Naylor et al., 2019), respectively). We evaluated the performance of our mixed annotation scheme using simulations where we vary the relative size of the two types of annotations. Namely, we did the following "drop-out" experiment: For each drop-out factor, f, between 0 to 0.9, we randomly selected an f-fraction of nuclei, and tested the effect of either removing the annotation completely (full dropout), or retaining the center location of the nuclei (partial dropout). For the rest (1-f)-fraction of nuclei full segmentation was retained. Figure 2 shows an example for ground-truth data (first-row panels) compared to corresponding predictions of model that was trained with no drop-outs, and to 50% partial dropout trained model (middle-row and bottom panels, respectively). As can be seen, the predictions of the partial-annotation model are very close to those of the fully-annotation model, and both are very close to the ground truth data. In Figure 2(e) we show relative segmentation accuracy (rAJI) for full dropout (red curve), and partial dropout (blue curve) as a function of drop-out factor. Those simulations demonstrate that while the model accuracy sharply deteriorate as expected with increased dropout factor (e.g., rAJI=0.72 at 0.7 dropout), retaining partial information allow for robust predictions (rAJI=0.96 at 0.7 dropout).

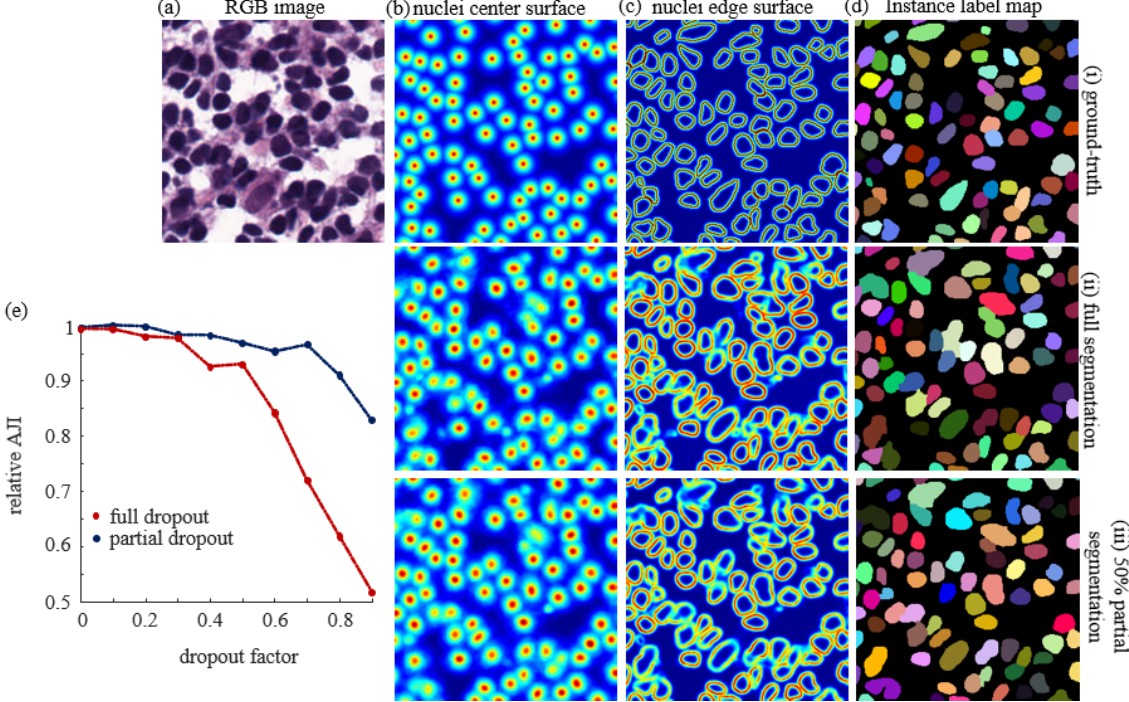

Figure 2: (a) RGB image, (b) nuclei center distance transform (c) edge distance transform (d) Instance segmentation; for (b)-(d): (i) ground-truth (ii) Network prediction for model trained with complete annotation (no drop out) (iii) Network prediction for 0.5 partial annotation trained model (e) evaluation of instance segmentation performance as a function of drop-out fraction

## 4. Summary

We presented a method that greatly minimizes the effort required to generate a training dataset for instance segmentation tasks, built upon two main components. (i) A novel instance segmentation ground-truth annotation scheme comprising a mixture of full annotation and partial annotation. (ii) A novel DL-based approach for instance segmentation that can be trained using such mixed annotation.

## Acknowledgments

We thank Anya Tsalenko and Steve Laderman for helpful comments on the abstract.

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
