# OpenReview forum: "Deep Learning Based Partial Annotation Framework for Instance Segmentation in Histopathology Images"
_MIDL.io/2019/Conference/Abstract — MIDL Abstract 2019_

### Official Review · AnonReviewer1 · 2019-04-30
**Interesting abstract**

**Rating:** 3
**Confidence:** 3

**Review:**

This paper proposes a method that can leverage a mixture of full annotation and partial annotation to train a segmeantion network. The model and the results are interesting and the abstract is written in a good manner.

---

### Official Review · AnonReviewer2 · 2019-05-01
**.**

**Rating:** 4
**Confidence:** 3

**Review:**

The paper deals with a very important problem in medical image analysis. Their method uses 30% of fully-segmented nuclei, and weakly segemented data, to achieve almost the same level of performance as having 100% fully-segmented nuclei.

---

### Decision · Program_Chairs · 2019-05-06
**Acceptance Decision**

Accept